# Local Linear Convergence of Forward–Backward under Partial Smoothness

**Jingwei Liang and Jalal M. Fadili**
GREYC, CNRS-ENSICAEN-Univ. Caen
{Jingwei.Liang,Jalal.Fadili}@greyc.ensicaen.fr

**Gabriel Peyré**
CEREMADE, CNRS-Univ. Paris-Dauphine
Gabriel.Peyre@ceremade.dauphine.fr

## Abstract

In this paper, we consider the Forward–Backward proximal splitting algorithm to minimize the sum of two proper closed convex functions, one of which having a Lipschitz continuous gradient and the other being partly smooth relative to an active manifold $\mathcal{M}$. We propose a generic framework under which we show that the Forward–Backward (i) correctly identifies the active manifold $\mathcal{M}$ in a finite number of iterations, and then (ii) enters a local linear convergence regime that we characterize precisely. This gives a grounded and unified explanation to the typical behaviour that has been observed numerically for many problems encompassed in our framework, including the Lasso, the group Lasso, the fused Lasso and the nuclear norm regularization to name a few. These results may have numerous applications including in signal/image processing processing, sparse recovery and machine learning.

## 1 Introduction

### 1.1 Problem statement

Convex optimization has become ubiquitous in most quantitative disciplines of science. A common trend in modern science is the increase in size of datasets, which drives the need for more efficient optimization methods. Our goal is the generic minimization of composite functions of the form

$$\min_{x \in \mathbb{R}^n} \big\{ \Phi(x) = F(x) + J(x) \big\}, \tag{1.1}$$

where

- **(A.1)** $J : \mathbb{R}^n \to \mathbb{R} \cup \{+\infty\}$ is a proper, closed and convex function;
- **(A.2)** $F$ is a convex and $C^{1,1}(\mathbb{R}^n)$ function whose gradient is $\beta$–Lipschitz continuous;
- **(A.3)** $\operatorname{Argmin} \Phi \neq \emptyset$.

The class of problems (1.1) covers many popular non-smooth convex optimization problems encountered in various fields throughout science and engineering, including signal/image processing, machine learning and classification. For instance, taking $F = \frac{1}{2\lambda}\|y - A \cdot \|^2$ for some $A \in \mathbb{R}^{m \times n}$ and $\lambda > 0$, we recover the Lasso problem when $J = \| \cdot \|_1$, the group Lasso for $J = \| \cdot \|_{1,2}$, the fused Lasso for $J = \|D^* \cdot \|_1$ with $D = [D_{\mathrm{DIF}}, \epsilon \mathrm{Id}]$ and $D_{\mathrm{DIF}}$ is the finite difference operator, anti-sparsity regularization when $J = \| \cdot \|_\infty$, and nuclear norm regularization when $J = \| \cdot \|_*$.

The standard (non relaxed) version of Forward–Backward (FB) splitting algorithm [3] for solving (1.1) updates to a new iterate $x_{k+1}$ according to

$$x_{k+1} = \mathrm{prox}_{\gamma_k J}\big(x_k - \gamma_k \nabla F(x_k)\big), \tag{1.2}$$

starting from any point $x_0 \in \mathbb{R}^n$, where $0 < \underline{\gamma} \le \gamma_k \le \overline{\gamma} < 2/\beta$. Recall that the proximity operator is defined, for $\gamma > 0$, as

$$\mathrm{prox}_{\gamma J}(x) = \mathrm{argmin}_{z \in \mathbb{R}^n} \tfrac{1}{2\gamma}\|z - x\|^2 + J(z).$$

## 1.2 Contributions

In this paper, we present a unified local linear convergence analysis for the FB algorithm to solve (1.1) when $J$ is in addition partly smooth relative to a manifold $\mathcal{M}$ (see Definition 2.1 for details). The class of partly smooth functions is very large and encompasses all previously discussed examples as special cases. More precisely, we first show that FB has a finite identification property, meaning that after a finite number of iterations, say $K$, all iterates obey $x_k \in \mathcal{M}$ for $k \ge K$. Exploiting this property, we then show that after such a large enough number of iterations, $x_k$ converges locally linearly. We characterize this regime and the rates precisely depending on the structure of the active manifold $\mathcal{M}$. In general, $x_k$ converges locally $Q$-linearly, and when $\mathcal{M}$ is an linear subspace, the convergence becomes $R$-linear. Several experimental results on some of the problems discussed above are provided to support our theoretical findings.

## 1.3 Related work

Finite support identification and local $R$-linear convergence of FB to solve the Lasso problem, though in infinite-dimensional setting, is established in [4] under either a very restrictive injectivity assumption, or a non-degeneracy assumption which is a specialization of ours (see (3.1)) to the $\ell_1$ norm. A similar result is proved in [13], for $F$ being a smooth convex and locally $C^2$ function and $J$ the $\ell_1$ norm, under restricted injectivity and non-degeneracy assumptions. The $\ell_1$ norm is a partly smooth function and hence covered by our results. [1] proved $Q$-linear convergence of FB to solve (1.1) for $F$ satisfying restricted smoothness and strong convexity assumptions, and $J$ being a so-called convex decomposable regularizer. Again, the latter is a small subclass of partly smooth functions, and their result is then covered by ours. For example, our framework covers the total variation (TV) semi-norm and $\ell_\infty$-norm regularizers which are not decomposable.

In [15, 16], the authors have shown finite identitification of active manifolds associated to partly smooth functions for various algorithms, including the (sub)gradient projection method, Newton-like methods, the proximal point algorithm. Their work extends that of e.g. [28] on identifiable surfaces from the convex case to a general non-smooth setting. Using these results, [14] considered the algorithm [25] to solve (1.1) where $J$ is partly smooth, but not necessarily convex and $F$ is $C^2(\mathbb{R}^n)$, and proved finite identitification of the active manifold. However, the convergence rate remains an open problem in all these works.

## 1.4 Notations

Suppose $\mathcal{M} \subset \mathbb{R}^n$ is a $C^2$-manifold around $x \in \mathbb{R}^n$, denote $\mathcal{T}_{\mathcal{M}}(x)$ the tangent space of $\mathcal{M}$ at $x \in \mathbb{R}^n$. The tangent model subspace is defined as

$$T_x = \mathrm{Lin}\big(\partial J(x)\big)^\perp,$$

where $\mathrm{Lin}(\mathcal{C})$ is the linear hull of the convex set $\mathcal{C} \subset \mathbb{R}^n$. For a linear subspace $V$, we denote $\mathrm{P}_V$ the orthogonal projector onto $V$ and for a matrix $A \in \mathbb{R}^{m \times n}$, $A_V = A \circ \mathrm{P}_V$. Define the generalized sign vector

$$e_x = \mathrm{P}_{T_x}\big(\partial J(x)\big).$$

For a convex set $\mathcal{C} \subset \mathbb{R}^n$, $\mathrm{ri}(\mathcal{C})$ denotes its relative interior, i.e. the interior relative to its affine hull.

# 2 Partial smoothness

In addition to (**A.1**), our central assumption is that $J$ is a partly smooth function. Partial smoothness of functions is originally defined in [17]. Our definition hereafter specializes it to the case of proper closed convex functions.

**Definition 2.1.** Let $J$ be a proper closed convex function such that $\partial J(x) \neq \emptyset$. $J$ is *partly smooth at $x$ relative to a set $\mathcal{M}$ containing $x$* if

(1) (Smoothness) $\mathcal{M}$ is a $C^2$-manifold around $x$ and $J$ restricted to $\mathcal{M}$ is $C^2$ around $x$.

(2) (Sharpness) The tangent space $\mathcal{T}_{\mathcal{M}}(x)$ is $T_x$.

(3) (Continuity) The set–valued mapping $\partial J$ is continuous at $x$ relative to $\mathcal{M}$.

In the following, the class of partly smooth functions at $x$ relative to $\mathcal{M}$ is denoted as $\mathrm{PS}_x(\mathcal{M})$. When $\mathcal{M}$ is an affine manifold, then $\mathcal{M} = x + T_x$, and we denote this subclass as $\mathrm{PSA}_x(x + T_x)$. When $\mathcal{M}$ is a linear manifold, then $\mathcal{M} = T_x$, and we denote this subclass as $\mathrm{PSL}_x(T_x)$.

Capitalizing on the results of [17], it can be shown that under mild transversality assumptions, the set of continuous convex partly smooth functions is closed under addition and pre-composition by a linear operator. Moreover, absolutely permutation-invariant convex and partly smooth functions of the singular values of a real matrix, i.e. spectral functions, are convex and partly smooth spectral functions of the matrix [10].

It then follows that all the examples discussed in Section 1, including $\ell_1, \ell_1 - \ell_2, \ell_\infty$, TV and nuclear norm regularizers, are partly smooth. In fact, the nuclear norm is partly smooth at a matrix $x$ relative to the manifold $\mathcal{M} = \{x' : \mathrm{rank}(x') = \mathrm{rank}(x)\}$. The first three regularizers are all part of the class $\mathrm{PSL}_x(T_x)$, see Section 4 and [27] for details.

We now define a subclass of partly smooth functions where the active manifold is actually a subspace and the generalized sign vector $e_x$ is locally constant.

**Definition 2.2.** $J$ belongs to the class $\mathrm{PSS}_x(T_x)$ if and only if $J \in \mathrm{PSA}_x(x + T_x)$ or $J \in \mathrm{PSL}_x(T_x)$ and $e_x$ is constant near $x$, i.e. there exists a neighbourhood $U$ of $x$ such that $\forall x' \in T_x \cap U$

$$e_{x'} = e_x.$$

A typical family of functions that comply with this definition is that of partly polyhedral functions [26, Section 6.5], which includes the $\ell_1$ and $\ell_\infty$ norms, and the TV semi-norm.

# 3 Local linear convergence of the FB method

In this section, we state our main result on finite identification and local linear convergence of FB.

**Theorem 3.1.** *Assume that (A.1)-(A.3) hold. Suppose that the FB scheme is used to create a sequence $x_k$ which converges to $x^\star \in \mathrm{Argmin}\,\Phi$ such that $J \in \mathrm{PS}_{x^\star}(\mathcal{M}_{x^\star})$, $F$ is $C^2$ near $x^\star$ and*

$$-\nabla F(x^\star) \in \mathrm{ri}\left(\partial J(x^\star)\right). \tag{3.1}$$

*Then we have the following holds,*

(1) *The FB scheme (1.2) has the finite identification property, i.e. there exists $K \geq 0$, such that for all $k \geq K$, $x_k \in \mathcal{M}_{x^\star}$.*

(2) *Suppose moreover that $\exists \alpha > 0$ such that*
$$\mathrm{P}_T \nabla^2 F(x^\star) \mathrm{P}_T \succeq \alpha \mathrm{Id}, \tag{3.2}$$
*where $T := T_{x^\star}$. Then for all $k \geq K$, the following holds.*

(i) *Q-linear convergence: if $0 < \underline{\gamma} \leq \gamma_k \leq \bar{\gamma} < \min\left(2\alpha\beta^{-2}, 2\beta^{-1}\right)$, then given any $1 > \rho > \widetilde{\rho}$,*
$$\|x_{k+1} - x^\star\| \leq \rho \|x_k - x^\star\|,$$
*where $\widetilde{\rho}^2 = \max\left\{q(\underline{\gamma}), q(\bar{\gamma})\right\} \in [0, 1[$ and $q(\gamma) = 1 - 2\alpha\gamma + \beta^2\gamma^2$.*

(ii) *R-linear convergence: if $J \in \mathrm{PSA}_{x^\star}(x^\star + T)$ or $J \in \mathrm{PSL}_{x^\star}(T)$, then for $0 < \underline{\gamma} \le \gamma_k \le \bar{\gamma} < \min\left(2\alpha\nu^{-2}, 2\beta^{-1}\right)$, where $\nu \le \beta$ is the Lipzchitz constant of $\mathrm{P}_T\nabla F\mathrm{P}_T$, then*

$$\|x_{k+1} - x^\star\| \le \rho_k \|x_k - x^\star\|,$$

*where $\rho_k^2 = 1 - 2\alpha\gamma_k + \nu^2\gamma_k^2 \in [0,1[$. Moreover, if $\frac{\alpha}{\nu^2} \le \bar{\gamma}$ and set $\gamma_k \equiv \frac{\alpha}{\nu^2}$, then the optimal linear rate can be achieved is*

$$\rho^* = \sqrt{1 - \alpha^2/\nu^2}.$$

**Remark 3.2.**
- The non-degeneracy assumption in (3.1) can be viewed as a geometric generalization of strict complementarity of non-linear programming. Building on the arguments of [16], it turns out that it is almost a necessary condition for finite identification of $\mathcal{M}_{x^\star}$.

- Under the non-degeneracy and local strong convexity assumptions (3.1)-(3.2), one can actually show that $x^\star$ is unique by extending the reasoning in [26].

- For $F = G \circ A$, where $G$ satisfies (**A.2**), assumption (3.2) and the constant $\alpha$ can be restated in terms of local strong convexity of $G$ and restricted injectivity of $A$ on $T$, i.e. $\mathrm{Ker}(A) \cap T = \{0\}$.

- When $F = \frac{1}{2}\|y - A \cdot\|^2$, not only the minimizer $x^\star$ is unique, but also the rates in Theorem 3.1 can be refined further as the gradient operator $\nabla F$ becomes linear.

- Partial smoothness guarantees that $x_k$ arrives the active manifold in finite time, hence raising the hope of acceleration using second-order information. For instance, one can think of turning to geometric methods along the manifold $\mathcal{M}_{x^\star}$, where faster convergence rates can be achieved. This is also the motivation behind the work of e.g. [19].

When $J \in \mathrm{PSS}_{x^\star}(T)$, it turns out that the restricted convexity assumption (3.2) of Theorem 3.1 can be removed in some cases, but at the price of less sharp rates.

**Theorem 3.3.** *Assume that (**A.1**)-(**A.3**) hold. For $x^\star \in \mathrm{Argmin}\,\Phi$, suppose that $J \in \mathrm{PSS}_{x^\star}(T_{x^\star})$, (3.1) is fulfilled, and there exists a subspace $V$ such that $\mathrm{Ker}\left(\mathrm{P}_T\nabla^2 F(x)\mathrm{P}_T\right) = V$ for any $x \in \mathbb{B}_\epsilon(x^\star)$, $\epsilon > 0$. Let the FB scheme be used to create a sequence $x_k$ that converges to $x^\star$ with $0 < \underline{\gamma} \le \gamma_k \le \bar{\gamma} < \min\left(2\alpha\beta^{-2}, 2\beta^{-1}\right)$, where $\alpha > 0$ (see the proof). Then there exists a constant $C > 0$ and $\rho \in [0,1[$ such that for all $k$ large enough*

$$\|x_k - x^\star\| \le C\rho^k.$$

A typical example where this result applies is for $F = G \circ A$ with $G$ locally strongly convex, in which case $V = \mathrm{Ker}(A_T)$.

# 4 Numerical experiments

In this section, we describe some examples to demonstrate the applicability of our results. More precisely, we consider solving

$$\min_{x \in \mathbb{R}^n} \tfrac{1}{2}\|y - Ax\|^2 + \lambda J(x) \tag{4.1}$$

where $y \in \mathbb{R}^m$ is the observation, $A : \mathbb{R}^n \to \mathbb{R}^m$, $\lambda$ is the tradeoff parameter, and $J$ is either the $\ell_1$-norm, the $\ell_\infty$-norm, the $\ell_1 - \ell_2$-norm, the TV semi-norm or the nuclear norm.

**Example 4.1 ($\ell_1$-norm).** For $x \in \mathbb{R}^n$, the sparsity promoting $\ell_1$-norm [8, 23] is

$$J(x) = \sum_{i=1}^{n} |x_i|.$$

It can verified that $J$ is a polyhedral norm, and thus $J \in \mathrm{PSS}_x(T_x)$ for the model subspace

$$\mathcal{M} = T_x = \big\{ u \in \mathbb{R}^n : \ \mathrm{supp}(u) \subseteq \mathrm{supp}(x) \big\}, \ \text{ and } \ e_x = \mathrm{sign}(x).$$

The proximity operator of the $\ell_1$-norm is given by a simple soft-thresholding.

**Example 4.2 ($\ell_1 - \ell_2$-norm).** The $\ell_1 - \ell_2$-norm is usually used to promote group-structured sparsity [29]. Let the support of $x \in \mathbb{R}^n$ be divided into non-overlapping blocks $\mathcal{B}$ such that $\bigcup_{b \in \mathcal{B}} b = \{1, \ldots, n\}$. The $\ell_1 - \ell_2$-norm is given by

$$J(x) = \|x\|_{\mathcal{B}} = \sum_{b \in \mathcal{B}} \|x_b\|,$$

where $x_b = (x_i)_{i \in b} \in \mathbb{R}^{|b|}$. $\| \cdot \|_{\mathcal{B}}$ in general is not polyhedral, yet partly smooth relative to the linear manifold

$$\mathcal{M} = T_x = \{u \in \mathbb{R}^n : \operatorname{supp}_{\mathcal{B}}(u) \subseteq \operatorname{supp}_{\mathcal{B}}(x)\}, \quad \text{and} \quad e_x = (\mathcal{N}(x_b))_{b \in \mathcal{B}},$$

where $\operatorname{supp}_{\mathcal{B}}(x) = \bigcup \{b : x_b \neq 0\}, \mathcal{N}(x) = x/\|x\|$ and $\mathcal{N}(0) = 0$. The proximity operator of the $\ell_1 - \ell_2$ norm is given by a simple block soft-thresholding.

**Example 4.3 (Total Variation).** As stated in the introduction, partial smoothness is preserved under pre-composition by a linear operator. Let $J_0$ be a closed convex function and $D$ is a linear operator. Popular examples are the TV semi-norm in which case $J_0 = \| \cdot \|_1$ and $D^* = D_{\mathrm{DIF}}$ is a finite difference approximation of the derivative [22], or the fused Lasso for $D = [D_{\mathrm{DIF}}, \epsilon \mathrm{Id}]$ [24].

If $J_0 \in \mathrm{PS}_{D^* x}(\mathcal{M}_0)$, then it is shown in [17, Theorem 4.2] that under an appropriate transversality condition, $J \in \mathrm{PS}_x(\mathcal{M})$ where

$$\mathcal{M} = \{u \in \mathbb{R}^n : D^* u \in \mathcal{M}_0\}.$$

In particular, for the case of the TV semi-norm, we have $J \in \mathrm{PSS}_x(T_x)$ with

$$\mathcal{M} = T_x = \{u \in \mathbb{R}^n : \operatorname{supp}(D^* u) \subseteq I\} \quad \text{and} \quad e_x = \mathrm{P}_{T_x} D \mathrm{sign}(D^* x)$$

where $I = \operatorname{supp}(D^* x)$. The proximity operator for the 1D TV, though not available in closed form, can be obtained efficiently using either the taut string algorithm [11] or the graph cuts [7].

**Example 4.4 (Nuclear norm).** Low-rank is the spectral extension of vector sparsity to matrix-valued data $x \in \mathbb{R}^{n_1 \times n_2}$, i.e. imposing the sparsity on the singular values of $x$. Let $x = U \Lambda_x V^*$ a reduced singular value decomposition (SVD) of $x$. The nuclear norm of a $x$ is defined as

$$J(x) = \|x\|_* = \sum_{i=1}^{r} (\Lambda_x)_i,$$

where $\mathrm{rank}(x) = r$. It has been used for instance as SDP convex relaxation for many problems including in machine learning [2, 12], matrix completion [21, 5] and phase retrieval [6].

It can be shown that the nuclear norm is partly smooth relative to the manifold [18, Example 2]

$$\mathcal{M} = \{z \in \mathbb{R}^{n_1 \times n_2} : \mathrm{rank}(z) = r\}.$$

The tangent space to $\mathcal{M}$ at $x$ and $e_x$ are given by

$$\mathcal{T}_{\mathcal{M}}(x) = \{z \in \mathbb{R}^{n_1 \times n_2} : z = U L^* + M V^*, \forall L \in \mathbb{R}^{n_2 \times r}, M \in \mathbb{R}^{n_1 \times r}\}, \quad \text{and} \quad e_x = U V^*.$$

The proximity operator of the nuclear norm is just soft–thresholding applied to the singular values.

**Recovery from random measurements**     In these examples, the forward observation model is

$$y = A x_0 + \varepsilon, \quad \varepsilon \sim \mathcal{N}(0, \delta^2), \tag{4.2}$$

where $A \in \mathbb{R}^{m \times n}$ is generated uniformly at random from the Gaussian ensemble with i.i.d. zero-mean and unit variance entries. The tested experimental settings are

   (a) $\ell_1$**-norm** $m = 48$ and $n = 128$, $x_0$ is 8-sparse;

   (b) **Total Variation** $m = 48$ and $n = 128$, $(D_{\mathrm{DIF}} x_0)$ is 8-sparse;

   (c) $\ell_\infty$**-norm** $m = 123$ and $n = 128$, $x_0$ has 10 saturating entries;

   (d) $\ell_1 - \ell_2$**-norm** $m = 48$ and $n = 128$, $x_0$ has 2 non-zero blocks of size 4;

   (e) **Nuclear norm** $m = 1425$ and $n = 2500$, $x_0 \in \mathbb{R}^{50 \times 50}$ and $\mathrm{rank}(x_0) = 5$.

The number of measurements is chosen sufficiently large, $\delta$ small enough and $\lambda$ of the order of $\delta$ so that [27, Theorem 1] applies, yielding that the minimizer of (4.1) is unique and verifies the non-degeneracy and restricted strong convexity assumptions (3.1)-(3.2).

The convergence profile of $\|x_k - x^\star\|$ are depicted in Figure 1(a)-(e). Only local curves after activity identification are shown. For $\ell_1$, TV and $\ell_\infty$, the predicted rate coincides exactly with the observed one. This is because these regularizers are all partly polyhedral gauges, and the data fidelity is quadratic, hence making the predictions of Theorem 3.1(ii) exact. For the $\ell_1 - \ell_2$-norm, although its active manifold is still a subspace, the generalized sign vector $e_k$ is not locally constant, which entails that the the predicted rate of Theorem 3.1(ii) slightly overestimates the observed one. For the nuclear norm, whose active manifold is not linear. Thus Theorem 3.1(i) applies, and the observed and predicted rates are again close.

**TV deconvolution** In this image processing example, $y$ is a degraded image generated according to the same forward model as (4.1), but now $A$ is a convolution with a Gaussian kernel. The anisotropic TV regularizer is used. The convergence profile is shown in Figure 1(f). Assumptions (3.1)-(3.2) are checked a posteriori. This together with the fact that the anisotropic TV is polyhedral justifies that the predicted rate is again exact.

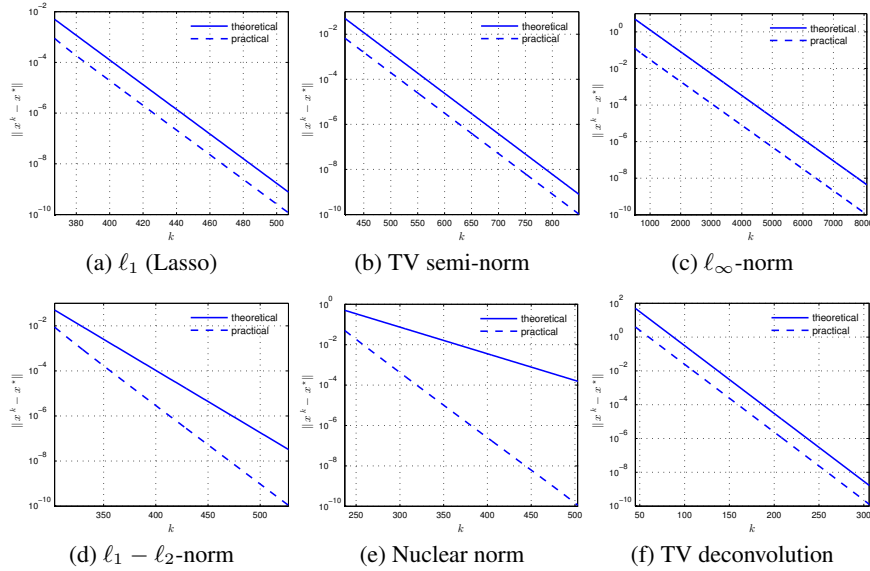

(a) $\ell_1$ (Lasso)    (b) TV semi-norm    (c) $\ell_\infty$-norm

(d) $\ell_1 - \ell_2$-norm    (e) Nuclear norm    (f) TV deconvolution

Figure 1: Observed and predicted local convergence profiles of the FB method (1.2) in terms of $\|x_k - x^\star\|$ for different types of partly smooth functions. (a) $\ell_1$-norm; (b) TV semi-norm; (c) $\ell_\infty$-norm; (d) $\ell_1 - \ell_2$-norm; (e) Nuclear norm; (f) TV deconvolution.

# 5  Proofs

**Lemma 5.1.** *Suppose that $J \in \mathrm{PS}_x(\mathcal{M})$. Then for any $x' \in \mathcal{M} \cap U$, where $U$ is a neighbourhood of $x$, the projector $\mathrm{P}_{\mathcal{M}}(x')$ is uniquely valued and $C^1$ around $x$, and thus*

$$x' - x = \mathrm{P}_{T_x}(x' - x) + o(\|x' - x\|).$$

*If $J \in \mathrm{PSA}_x(x + T_x)$ or $J \in \mathrm{PSL}_x(T_x)$, then*

$$x' - x = \mathrm{P}_{T_x}(x' - x).$$

**Proof.** Partial smoothness implies that $\mathcal{M}$ is a $C^2$–manifold around $x$, then $\mathrm{P}_{\mathcal{M}}(x')$ is uniquely valued [20] and moreover $C^1$ near $x$ [18, Lemma 4]. Thus, continuous differentiability shows

$$x' - x = \mathrm{P}_{\mathcal{M}}(x') - \mathrm{P}_{\mathcal{M}}(x) = \mathrm{DP}_{\mathcal{M}}(x)(x - x') + o(\|x - x'\|).$$

where $\mathrm{DP}_{\mathcal{M}}(x)$ is the derivative of $\mathrm{P}_{\mathcal{M}}$ at $x$. By virtue of [18, Lemma 4] and the sharpness property of $J$, this derivative is given by

$$\mathrm{DP}_{\mathcal{M}}(x) = \mathrm{P}_{\mathcal{T}_{\mathcal{M}}(x)} = \mathrm{P}_{T_x},$$

The case where $\mathcal{M}$ is affine or linear is immediate. This conlcudes the proof. □

**Proof of Theorem 3.1.**

1. Classical convergence results of the FB scheme, e.g. [9], show that $x_k$ converges to some $x^\star \in$ Argmin $\Phi \neq \emptyset$ by assumption (**A.3**). Assumptions (**A.1**)-(**A.2**) entail that (3.1) is equivalent to $0 \in \mathrm{ri}\,\partial\big(\Phi(x^\star)\big)$. Since $F \in C^2$ around $x^\star$, the smooth perturbation rule of partly smooth functions [17, Corollary 4.7] ensures that $\Phi \in \mathrm{PS}_{x^\star}(\mathcal{M})$. By definition of $x_{k+1}$, we have

$$\tfrac{1}{\gamma_k}\big(G_k(x_k) - G_k(x_{k+1})\big) \in \partial\Phi(x_{k+1}).$$

   where $G_k = \big(\mathrm{Id} - \gamma_k \nabla F\big)$. By Baillon-Haddad theorem, $G_k$ is non-expansive, hence

$$\mathrm{dist}\big(0, \partial\Phi(x_{k+1})\big) \leq \tfrac{1}{\gamma_k}\|G_k(x_k) - G_k(x_{k+1})\| \leq \tfrac{1}{\gamma_k}\|x_k - x_{k+1}\|.$$

   Since $\liminf \gamma_k = \underline{\gamma} > 0$, we obtain $\mathrm{dist}\big(0, \partial\Phi(x_{k+1})\big) \to 0$. Owing to assumptions (**A.1**)-(**A.2**), $\Phi$ is subdifferentially continuous and thus $\Phi(x_k) \to \Phi(x^\star)$. Altogether, this shows that the conditions of [15, Theorem 5.3] are fulfilled, whence the claim follows.

2. Take $K > 0$ sufficiently large such that for all $k \geq K$, $x_k \in \mathcal{M}_{x^\star}$ and $x_k \in \mathbb{B}_\epsilon(x^\star)$.

   (i) Since $\mathrm{prox}_{\gamma_k J}$ is firmly non-expansive, hence non-expansive, we have

$$\|x_{k+1} - x^\star\| = \|\mathrm{prox}_{\gamma_k J} G_k x_k - \mathrm{prox}_{\gamma_k J} G_k x^\star\| \leq \|G_k x_k - G_k x^\star\|. \qquad (5.1)$$

   By virtue of Lemma 5.1, we have $x_k - x^\star = \mathrm{P}_T(x_k - x^\star) + o(\|x_k - x^\star\|)$. This, together with local $C^2$ smoothness of $F$ and Lipschitz continuity of $\nabla F$ entails

$$\langle x_k - x^\star, \nabla F(x_k) - \nabla F(x^\star)\rangle = \int_0^1 \langle x_k - x^\star, \nabla^2 F(x^\star + t(x_k - x^\star))(x_k - x^\star)\rangle dt$$

$$= \int_0^1 \langle \mathrm{P}_T(x_k - x^\star), \nabla^2 F(x^\star + t(x_k - x^\star))\mathrm{P}_T(x_k - x^\star)\rangle dt + o\big(\|x_k - x^\star\|^2\big)$$

$$\geq \alpha\|x_k - x^\star\|^2 + o\big(\|x_k - x^\star\|^2\big). \qquad (5.2)$$

   Since (3.2) holds and $\nabla^2 F(x)$ depends continuously on $x$, there exists $\epsilon > 0$ such that $\mathrm{P}_T \nabla^2 F(x) \mathrm{P}_T \succ \alpha\mathrm{Id}, \forall x \in \mathbb{B}_\epsilon(x^\star)$. Thus, classical development of the right hand side of (5.1) yields

$$\|x_{k+1} - x^\star\|^2 \leq \|G_k x_k - G_k x^\star\|^2 = \|(x_k - x^\star) - \gamma_k(\nabla F(x_k) - \nabla F(x^\star))\|^2$$

$$= \|x_k - x^\star\|^2 - 2\gamma_k\langle x_k - x^\star, \nabla F(x_k) - \nabla F(x^\star)\rangle + \gamma_k^2\|\nabla F(x_k) - \nabla F(x^\star)\|^2$$

$$\leq \|x_k - x^\star\|^2 - 2\gamma_k\alpha\|x_k - x^\star\|^2 + \gamma_k^2\beta^2\|x_k - x^\star\|^2 + o\big(\|x_k - x^\star\|^2\big)$$

$$= \big(1 - 2\alpha\gamma_k + \beta^2\gamma_k^2\big)\|x_k - x^\star\|^2 + o\big(\|x_k - x^\star\|^2\big). \qquad (5.3)$$

   Taking the lim sup in this inequality gives

$$\limsup_{k \to +\infty} \|x_{k+1} - x^\star\|^2/\|x_k - x^\star\|^2 \leq q(\gamma_k) = 1 - 2\alpha\gamma_k + \beta^2\gamma_k^2. \qquad (5.4)$$

   It is clear that for $0 < \underline{\gamma} \leq \gamma \leq \bar{\gamma} < \min\big(2\alpha\beta^{-2}, 2\beta^{-1}\big)$, $q(\gamma) \in [0,1[$, and $q(\gamma) \leq \widetilde{\rho}^2 = \max\{q(\underline{\gamma}), q(\bar{\gamma})\}$. Inserting this in (5.4), and using classical arguments yields the result.

   (ii) We give the proof for $\mathcal{M} = T$, that for $\mathcal{M} = x^\star + T$ is similar. Since $x_k$ and $x^\star$ belong to $T$, from $x_{k+1} = \mathrm{prox}_{\gamma_k J}(G_k x_k)$ we have

$$G_k x_k - x_{k+1} \in \gamma_k \partial J(x_{k+1}) \Rightarrow x_{k+1} = \mathrm{P}_T\big(G_k x_k - \gamma_k \partial J(x_{k+1})\big) = \mathrm{P}_T G_k x_k - \gamma_k e_{k+1}.$$

   Similarily, we have $x^\star = \mathrm{P}_T G_k x^\star - \gamma_k e^\star$. We then arrive at

$$(x_{k+1} - x^\star) + \gamma_k(e_{k+1} - e^\star) = (x_k - x^\star) - \gamma_k\big(\mathrm{P}_T \nabla F(\mathrm{P}_T x_k) - \mathrm{P}_T \nabla F(\mathrm{P}_T x^\star)\big). \quad (5.5)$$

Moreover, maximal monotonicity of $\gamma_k \partial J$ gives

$$\|(x_{k+1} - x^\star) + \gamma_k(e_{k+1} - e^\star)\|^2$$
$$= \|x_{k+1} - x^\star\|^2 + 2\langle x_{k+1} - x^\star, \gamma_k(e_{k+1} - e^\star)\rangle + \gamma_k\|e_{k+1} - e^\star\|^2 \geq \|x_{k+1} - x^\star\|^2.$$

It is straightforward to see that now, (5.2) becomes

$$\langle x_k - x^\star, \mathrm{P}_T\nabla F(\mathrm{P}_T x_k) - \mathrm{P}_T\nabla F(\mathrm{P}_T x^\star)\rangle \geq \alpha\|x_k - x^\star\|^2.$$

Let $\nu$ be the Lipschitz constant of $\mathrm{P}_T\nabla F\mathrm{P}_T$. Obviously $\nu \leq \beta$. Developing $\|\mathrm{P}_T(G_k x_k - G_k x^\star)\|^2$ similarly to (5.3) we obtain

$$\|x_{k+1} - x^\star\|^2 \leq \left(1 - 2\alpha\gamma_k + \nu^2\gamma_k^2\right)\|x_k - x^\star\|^2 = \rho_k^2\|x_k - x^\star\|^2,$$

where $\rho_k \in [0, 1[$ for $0 < \underline{\gamma} \leq \gamma_k \leq \bar{\gamma} < \min\left(2\alpha/\nu^2, 2/\beta\right)$. $\rho_k$ is minimized at $\frac{\alpha}{\nu^2}$ with the proposed optimal rate whenever it obeys the given upper-bound. $\qquad\square$

**Proof of Theorem 3.3.** Arguing similarly to the proof of Theorem 3.1(ii), and using in addition that $e^\star = e_{x^\star}$ is locally constant, we get

$$x_{k+1} - x^\star = (x_k - x^\star) - \gamma_k\left(\mathrm{P}_T\nabla F(\mathrm{P}_T x_k) - \mathrm{P}_T\nabla F(\mathrm{P}_T x^\star)\right)$$
$$= (x_k - x^\star) - \gamma_k \int_0^1 \mathrm{P}_T\nabla^2 F(x^\star + t(x_k - x^\star))\mathrm{P}_T(x_k - x^\star)dt,$$

Denote $H_t = \mathrm{P}_T\nabla^2 F(x^\star + t(x_k - x^\star))\mathrm{P}_T \succeq 0$. Using that $H_t$ is self-adjoint, we have

$$\mathrm{P}_V x_{k+1} = \mathrm{P}_V x_k.$$

Since $x_k \to x^\star$, it follows that $\mathrm{P}_V x_k = \mathrm{P}_V x^\star$ for all $k$ sufficiently large. Observing that $x_k - x^\star = \mathrm{P}_{V^\perp}(x_k - x^\star)$ for all large $k$, we get

$$x_{k+1} - x^\star = x_k - x^\star - \gamma_k \int_0^1 \mathrm{P}_{V^\perp}H_t\mathrm{P}_{V^\perp}(x_k - x^\star)dt.$$

Observe that $V^\perp \subset T$. By definition, $B_t = H_t^{1/2}\mathrm{P}_{V^\perp}$ is injective, and therefore, $\exists\sigma > 0$ such that $\|B_t x\|^2 > \sigma\|x\|^2$ for all $x \neq 0$ and $t \in [0, 1]$. We then have

$$\|x_{k+1} - x^\star\|^2$$
$$= \|x_k - x^\star\|^2 - 2\gamma_k \int_0^1 \langle x_k - x^\star, B_t^T B_t(x_k - x^\star)\rangle dt + \gamma_k^2\|\mathrm{P}_{V^\perp}\mathrm{P}_T\left(\nabla F(x_k) - \nabla F(x^\star)\right)\|^2$$
$$= \|x_k - x^\star\|^2 - 2\gamma_k \int_0^1 \|B_t(x_k - x^\star)\|^2 dt + \gamma_k^2\|\mathrm{P}_{V^\perp}\mathrm{P}_T\|^2\|\nabla F(x_k) - \nabla F(x^\star)\|^2$$
$$= \|x_k - x^\star\|^2 - 2\gamma_k\sigma\|x_k - x^\star\|^2 + \gamma_k^2\|\mathrm{P}_T\mathrm{P}_{V^\perp}\|^2\|\nabla F(x_k) - \nabla F(x^\star)\|^2$$
$$\leq \|x_k - x^\star\|^2 - 2\gamma_k\sigma\|x_k - x^\star\|^2 + \gamma_k^2\beta^2\|\mathrm{P}_{V^\perp}\|^2\|\mathrm{P}_{V^\perp}(x_k - x^\star)\|^2$$
$$\leq \|x_k - x^\star\|^2 - 2\gamma_k\sigma\|x_k - x^\star\|^2 + \gamma_k^2\beta^2\|x_k - x^\star\|^2 = \rho_k^2\|x_k - x^\star\|^2.$$

It is easy to see again that $\rho_k \in [0, 1[$ whenever $0 < \underline{\gamma} \leq \gamma_k \leq \bar{\gamma} < \min\left(2\beta^{-1}, 2\sigma\beta^{-2}\right)$. $\qquad\square$

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
