[Reviews · NeurIPS 2014]

Submitted by Assigned_Reviewer_29

The FBS algorithm has been rediscovered in the last 10 years along with other splitting algorithms like the Douglas-Rachford algorithm. FBS is a simple algorithm to minimize a functional which consists of the sum of a smooth function and one for which the prox-operator can be solved efficiently. It first applies a gradient descent step (thus it is a first-order method) on the smooth part after which the prox-operator of the nonsmooth part is applied. This algorithm is used in many machine learning and imaging applications and a lot of research has been done to speed it up (FISTA, Barzilai-Borwein step sizes etc.).
Hence, analyzing the convergence properties of FBS which is the topic of the current paper is interesting for the NIPS community.

The main contribution of this paper is to make use of the concept of partial smoothness to derive convergence results given in Theorem 3.1. Although this idea was employed before, the results seem novel and interesting.

Generally, I think that the paper would have improved a lot if the authors put the convergence proof in the supplementary material and spend these two pages to discuss their results in more detail. For example, the results in Remark 3.2 are interesting but it should be made clear whether they can be found somewhere, are easy to obtain or their proof is too long to be included here. In the last sentence of the abstract, the authors mention the importance for applications but only Remark 3.4 follows this thought. A natural question is how one can make use of the finite identification property, i.e., if there is a way to identify the index K of part 2. of the proof and how the algorithm can be adapted once K is reached. One could at least experimentally analyze finite identification property. The plots provided here focus on the asymptotic behavior. The example problems considered in the paper are well-chosen since one clearly sees differences according to cases i) and ii) of the main theorem. However, showing maybe one real-world example would be beneficial. In addition, especially for readers new to the topic, it would be worthwhile to make the step from Theorem 3.1 to the examples easier by explicitly checking all conditions of the theorem, even ones which might be clear like (3.1). The last paragraph on p. 5 about the difference in the tightness of the convergence rate are very interesting and should be treated in broader detail. Is there a mathematical justification why exactly the bottom two convergence curves are less tight? (apart from the general observation on the properties of the manifold and the sign vectors)

Furthermore, an outlook on generalizations (if possible) of the results presented here would be good: Do the results generalize to an infinite dimensional setting? Are they transferable to other splitting methods like the Douglas-Rachford method? Can parts of the results be carried over to the non-convex setting? How about versions of FBS which employ speed-up strategies, e.g. FISTA?

Minor things:
- define polyhedral gauge functions
- p. 3, l. 128: define "generalized sign vector"
- p. 3, l. 127: typo: subsclass
- p. 5, l. 234 typo: taut string algorithm
Summary: The analysis of the FBS algorithm provided here generalizes existing results and shows new insights but it remains unclear how this can be used to improve current algorithms.

Submitted by Assigned_Reviewer_32

The paper generalizes previous work on the identification and asymptotic rate of lasso-like problems to a larger class. It seems to be a rather straightforward extension of [15-18] and [28] combined with known results on the forward-backward algorithm convergence rate.

The technical content appears rigorous and correct, and assumptions are well-stated. The standard examples in the sparse approximation literature are covered.

There are no major flaws I can see. The main weakness of the paper is that it could be considered an incremental improvement. In section 1.3, the authors should describe more clearly if the current paper is a natural extension of [15-18] or if it requires significantly new ideas. The proof in section 5 could also benefit from a few sentences describing the main idea and innovations.

In section 1.3, it says that the convergence rate is an open problem in all of these works. Once the finite identification happens, convergence rates follow easily, and I would think that previous papers have noticed this. For example, a related result (basically, the same algorithm but in a non-convex setting) from "Iterative Thresholding for Sparse Approximations" by Blumensath and Davies shows the same identification and then derives the linear rate.

The manifold properties remind me of the "decomposable" norms of Negahban, Ravikumar, Wainwright and Yu '12. These decomposable norms have even more structure. It would be interesting to see the connection.

The discussion of Fig 1 was confusing. The text says that it is evidence that the observed and predicted rates are consistent. By "consistent", do you mean that the observations do not contradict the predictions, or do you mean that the rates match? The rates do not match -- the slope in (d) of the observations is clearly much steeper than the slope of the theoretical bound. Can you explain this?

It was nice that the proof was in the main section of the paper and not the appendix.

Typo in section 1.4: "linear of" should be "linear hull of"
The references should capitalize words like "Riemannian" and "SQP" (e.g., [20])

Clarity of the paper is good.
Summary: This is a generalization of other known results. It's useful and correct, but a bit incremental.

Submitted by Assigned_Reviewer_40

This paper analyzes the locally linear convergence of forward-backward (F-B) splitting algorithms when the objective function is a sum of two convex functions, one having a Lipschitz-continuous gradient and the other beig partly smooth relative to a manifold. This is an interesting paper as it reveals that the convergence of F-B for many common sparse or low-rank minimization problems are actually locally linear convergent, although the objective function may not be strongly convex.

However, the paper has the following issues:
1. The paper requires many background knowledge and results that scatter across some papers, making the paper not very self-contained. Reading through the paper requires a lot of painful effort. It will be better if the authors provide some introductory materials and quote the basic results (e.g., Theorem 5.3 of [16]). The proofs in Section 5, which are easy, can be moved to supplementary material.
2. It is obvious that the active manifold M is dependent on x. Then what does the M in 1. of Theorem 3.1 mean? It is the manifold at x^*? Could the notation M be changed to M_x for resolving ambiguity?
3. In the proof of 1. Theorem 3.1, I don't see why applying the Baillon-Haddad theorem results in the non-expansiveness of G_k. It should result in the non-expansiveness of \partial \Phi(x_{k+1}).

Minor issue:
1. in Line 097, "Lin(C) is the linear of the convex set C" should be "Lin(C) is the linear span of the convex set C"
2. In line 128, why e_x defined in line 099 called "generalized sign vector"? It is not related to sign.
3. A lot of English errors, most notably the title, which should be "Locally Linear Convergence of Forward-Backward Splitting Algorithm under Partial Smoothness".
4. References [1] and [2] are identical.
Summary: An interesting paper to understand the locally linear convergence of F-B for some common sparse and low-rank minimization problems. But the paper is difficult to read due not non-self-containedness.
Author Feedback
Author rebuttal: We are glad that all the reviewers generally appreciated the significance of our contributions.

Some however raised the issue that the contribution is a bit incremental, and we respectfully disagree. While finite activity identification is marginally new, local linear convergence is an important contribution. We agree (with Reviewer 32) that it is an intuitive result: after all, once identification is obtained, everything boils down to minimizing a smooth function on a smooth manifold. But we strongly disagree that the proof is trivial and incremental: it requires some serious mathematics to prove that linear convergence indeed holds. Let us put it this way: even if it were incremental (and we argue it is not!), it is new and it is such a fundamental result (given the importance of the FB algorithm) that we think deserves to be published. This is also confirmed by the few papers in the literature which only handled very special regularizers.

Some reviewers suggested improvements to better explain the relevance of this result. We believe this is fairly easy to address in the final version of the paper, and propose how to achieve this in the detailed answers below.

*** Reviewer 29
Q1: the paper would have improved a lot [...]
A: Though this will reduce self-­containedness (Q1 of Reviewer 40), we partly agree with this, and we'll update the manuscript accordingly.

Q2: the results in Remark 3.2 are interesting [...]
A: These are simple results, we will add some more details to explain how they can be derived.

Q3: how to make use of the finite identification property [...]
A: Basically, we have devised some heuristics to empirically determine approximate values of K, which works well for the applications under study. However we feel it is somehow beyond the scope of the paper to discuss them in detail, we will add a remark about this and the potential applications.

Q4: the plots focus on the asymptotic behavior [...]
A: In all the considered examples, we have observed that the value of K closely matches the point where the error decay becomes linear. We will add a comment on this.

Q5: one real-world example [...]
A: Due to the lack of space, we have focused on simple examples that are easy to expose and understand. There is no difficulty applying the result to large-scale problems. In fact we already have examples on imaging and computer vision problems. We will add one in the supplementary material.

Q6: for readers new to the topic [...]
A: The setups considered in these examples have been designed so that these conditions always hold in practice with high probability on the design matrix. We will add more details about this and explain how to practically check the condition of l^1 and nuclear norm examples.

Q7: p. 5 about the difference in the tightness [...]
A: The difference in tightness corresponds to the fact that the considered regularizers (l1-l2 and nuclear norm) are not partly polyhedral, and the curvature of the level-sets induces this non-tightness. More precisely, when x->e_x is non-constant along T, the inequalities on Pages 7-8 are not tight. These inequalities are just equalities in the partly polyhedral case. We will add more comments on this.

Q8: an outlook on generalizations [...]
A: These are interesting questions that we are currently working on, and some of which we have already solved but are much beyond the scope of this paper. We will mention this in the conclusion.

*** Reviewer 32
Q1: if the current paper is a natural extension of [15-18] [...]
A: The finite activity identification is indeed an almost straightforward extension of these works. It is however *not* our main contribution. The proof of linear convergence, while it might appear very intuitive, is not straightforward, and requires some non-trivial proof. We will make this clearer in the "related work" section.

Q2: The proof in section 5 could also benefit [...]
A: We will add a paragraph on this.

Q3: when finite identification happens, convergence rates follow easily [...]
A: We respectfully disagree with this statement.

Q4: previous papers have noticed this
A: No, they did not, with the exception of the special cases considered in two papers that we discuss in the " related work" section.

Q5: For example, a related result [...]
A: This work uses a RIP-based proof, and relies strongly on the piecewise constant nature of l^0 norm, and is a very particular (non-convex) case. We feel it is unfair to use this particular case to qualify our result as incremental. A More related work is the one we cite and discuss in the " related work" section.

Q6: relation with the "decomposable" norms [...]
A: Decomposable norm is a tiny subclass of partly smooth functions as we argue in the "previous work" section. For instance, l^inf norm or TV-semi norm are not "decomposable". We will add some more comments about this.

Q7: discussion of Fig 1 was confusing [...]
A: We agree that the phrasing is a bit confusing, and will modify it.

Q8: The rates do not match [...]
A: See answer to Q7 of Reviewer 29.

**** Reviewer 40
Q1: The paper requires many background knowledge and results [...]
A: We agree and will add some introductory material by quoting existing monograph, e.g. Bauschke and Combettes's.

Q2: active manifold M is dependent on x [...]
A: The manifold M in Theorem 3.1 is the manifold associated to x^*, to clarify this, we will denote it as M_{x^*}.

Q3: why applying the Baillon-Haddad theorem [...]
A: The subdifferential partial Phi is a set-valued operator and *cannot* be non-expansive in the classical sense. It is indeed non-expansiveness of G_k that we need, and this is a consequence of Baillon-Haddad theorem which states that \nabla F is firmly non-expansive, and thus that G_k is \alpha_k-averaged, hence non-expansive, for the prescribed range of \gamma_k. We will clarify this.